# The Serum Brain-Derived Neurotrophic Factor Increases in Serotonin Reuptake Inhibitor Responders Patients with First-Episode, Drug-Naïve Major Depression

**DOI:** 10.3390/biomedicines11020584

**Published:** 2023-02-16

**Authors:** Reiji Yoshimura, Naomichi Okamoto, Enkmurun Chibaatar, Tomoya Natsuyama, Atsuko Ikenouchi

**Affiliations:** Department of Psychiatry, University of Occupational and Environmental Health, Iseigaoka, Yahatanishi-ku, Kitakyushu 8078555, Fukuoka, Japan

**Keywords:** brain-derived neurotrophic factor, serum, trajectory, major depression, first-episode, drug-naïve, Diagnostic and Statistical Manual of Mental Disorders Fifth Edition, Hamilton Rating Scale for Depression, paroxetine, escitalopram, duloxetine

## Abstract

Brain-derived neurotrophic factor (BDNF) is a growth factor synthesized in the cell bodies of neurons and glia, which affects neuronal maturation, the survival of nervous system, and synaptic plasticity. BDNF play an important role in the pathophysiology of major depression (MD). The serum BDNF levels changed over time, or with the improvement in depressive symptoms. However, the change of serum BDNF during pharmacotherapy remains obscure in MDD. In particular, the changes in serum BDNF associated with pharmacotherapy have not yet been fully elucidated. The present study aimed to compare the changes in serum BDNF concentrations in first-episode, drug-naive patients with MD treated with antidepressants between treatment-response and treatment-nonresponse groups. The study included 35 inpatients and outpatients composed of 15 males and 20 females aged 36.7 ± 6.8 years at the Department of Psychiatry of our University Hospital. All patients met the DSM-5 diagnostic criteria for MD. The antidepressants administered included paroxetine, duloxetine, and escitalopram. Severity of depressive state was assessed using the 17-item HAMD before and 8 weeks after drug administration. Responders were defined as those whose total HAMD scores at 8 weeks had decreased by 50% or more compared to those before drug administration, while non-responders were those whose total HAMD scores had decreased by less than 50%. Here we showed that serum BDNF levels were not significantly different at any point between the two groups. The responder group, but not the non-responder group, showed statistically significant changes in serum BDNF 0 and serum BDNF 8. The results suggest that the changes of serum BDNF might differ between the two groups. The measurement of serum BDNF has the potential to be a useful predictor of pharmacotherapy in patients with first-episode, drug-naïve MD.

## 1. Introduction

Major depression (MD) is common, with almost one in five people experiencing one episode at some point in their lifetime [1]. The symptoms of MD are related to structural and neurochemical deficits in the corticolimbic brain regions. The behavioral symptoms of MD are extensive, covering the emotional, motivational, cognitive, and physiological domains, and include anhedonia, aberrant reward-associated perception, and memory alterations [2]. The precise pathophysiology of MD, however, remains unknown. The understanding of the pathophysiology of MD has progressed, but the single model and its mechanisms cannot explain the entirety of MD. Monoamines [3], the abnormality of the hypothalamic-pituitary-adrenal axis [4], inflammation [5], neuroplasticity, neurogenesis, neurocircuit [6], or structural change including in the prefronto-striatal-limbic and fronto-parietal network, which is associated with cognitive impairment, [7,8,9] or reduced resilience to mitochondrial impairment [10] might be involved in the pathophysiology of MD.

BDNF plays a major role in neuronal growth and survival, functions as a neurotransmitter modulator, and contributes to neuronal plasticity. BDNF stimulates and regulates the growth of new neurons from neural stem cells (i.e., neurogenesis); BDNF has been linked to synaptic re-modeling, being able to both induce and be induced by long-term potentiation [11,12]. The human BDNF gene consists of 11 exons and 9 promoters [13]. Both BDNF proteins and mRNA have been detected in various brain regions, including the olfactory bulb, cerebral cortex, hippocampus, basal forebrain, mesencephalon, hypothalamus, brainstem, and spinal cord [14,15,16,17,18,19]. Therefore, it has been suggested that abnormalities in BDNF in the brain are associated with the pathogenesis of MD [20,21,22,23,24,25,26]. It has been reported that the BDNF val66met polymorphism affects the activity-dependent secretion of BDNF and human memory and hippocampal function. In short, BDNF gene val/met polymorphism is associated with human memory and hippocampal function, and val/met polymorphism exerts these effects by impacting intracellular trafficking and the activity-dependent secretion of BDNF [27]. Furthermore, the BDNF gene Val66Met polymorphism is associated with vulnerability to MD [28,29].

Several meta-analyses have shown that serum and plasma levels of BDNF were significantly decreased in patients with MD compared to healthy controls [30,31,32,33,34,35]. Similarly, in our previous work, we also reported that serum BDNF concentrations were significantly lower in untreated patients with MD than in healthy controls [34,35]. A meta-analysis showed that various antidepressant treatments increase serum and plasma BDNF concentrations in patients with MD [34,36,37,38,39]. We also previously found that paroxetine and milnacipran significantly increased serum BDNF concentrations after 8 weeks in untreated patients with MD in the treatment response group [36,37]. Although there have been reports of increased serum and plasma BDNF levels after antidepressant treatment in patients with MD [21,22], the findings of the relationship between changes in serum or plasma BDNF levels and response to pharmacotherapy, however controversial, cannot be fully elucidated. Response to duloxetine was associated with a higher baseline serum BDNF level and greater reduction of the Hamilton Rating Scale for Depression (HAMD) scores for MD [40]. The absence of an early increase of serum BDNF levels in conjunction with early non-response to antidepressants can be a highly specific peripheral biomarker predictive for treatment failure of pharmacotherapy in MD [41]. Alternatively, the combination of an early increase (day 7) of plasma BDNF with early reduction of the HAMD scores could be a useful predictive marker for pharmaco-treatment in MD [42]. A decrease of serum BDNF levels at week 2 of selective serotonin reuptake inhibitor (SSRI) treatment might be associated with later SSRI response in adolescents with MD [43]. Pretreatment serum BDNF levels have been reported to be correlated with antidepressant responses, and responders to treatment improvement in severity of MD had higher pretreatment serum BDNF levels than did non-responders [40,43]. In short, the results of the time course of serum/plasma BDNF levels and response to antidepressants were not consistent and remain obscure. Moreover, there are no reports of detailed observations of the time course of serum or plasma BDNF levels and the response to antidepressants in MD patients. 

Considering the above results and evidence, we sought to compare the changes of serum BDNF concentrations between the treatment-response and treatment-nonresponse groups in first-episode, drug-naive patients with MD treated with antidepressants. In other words, the aim of this study is to identify the changes in serum BDNF as an indicator of drug responsiveness.

## 2. Patients and Methods

### 2.1. Study Population

This study included 35 inpatients and outpatients, composed of 15 males and 20 females with a mean age of 36.7 ± 6.8 years, at the Department of Psychiatry, Occupational and Environmental Medicine Hospital. The detailed demographics of the patients are described in Table 1. All patients met the Diagnostic and Statistical Manual of Mental Disorders Fifth Edition (DSM-5) [44] diagnostic criteria for MD and were first-episode and drug-naive. 

### 2.2. Treatment and Assessment of Depression

The antidepressants administered were paroxetine in 15 patients (average dose: 32.0 ± 9.7 mg/day), duloxetine in 11 patients (average dose: 49.0 ± 9.9 mg/day), and escitalopram in 9 patients (average dose: 18.8 ± 3.1 mg/day) (Table 2). Severity of depressive state was assessed using the 17-item Hamilton Depression Rating Scale (HAMD) [45] before and 8 weeks after drug administration. Responders were defined as those whose total HAMD scores at 8 weeks had decreased by 50% or more compared to those before drug administration, while non-responders were those whose total HAMD scores had decreased by less than 50%. 

### 2.3. Blood Collection and Measurement of Serum BDNF

Blood samples were collected at 8–10 a.m. before drug administration; after 2, 4, and 8 weeks; before meals; and after a 30-min rest (Figure 1). The serum was separated, and the serum BDNF concentration was measured by enzyme-linked immunosorbent assay (ELISA) according to a previously reported method [46]. In brief, 96-well microplates were coated with anti-BDNF monoclonal antibodies and incubated at 4 °C for 18 h. The plates were incubated in a blocking buffer for 1 h at room temperature. The samples were diluted with an assay buffer by 100-times and BDNF standards were maintained at room temperature under conditions of horizontal shaking for 2 h, followed by washing with the appropriate washing buffer. The plates were incubated with antihuman BDNF polyclonal antibodies at room temperature for 2 h and washed with the washing buffer. The plates were then incubated with anti-IgY antibody conjugated to horseradish peroxidase for 1 h at room temperature and incubated in peroxidase substrate and tetramethylbenzidine solution to induce a color reaction. The reaction was stopped with 1 mol/L hydrochloric acid. The absorbance at 450 nm was measured with an Emax automated microplate reader. Measurements were performed in duplicates. The standard curve was linear from 0.5 ng/mL to 50 ng/mL, and the detection limit was 5 ng/mL. The cross-reactivity to related neurotrophies (NT-3, NT-4, NGF) was less than 3%. Intra- and inter-assay coefficients of variation were about 5% and 7%, respectively. The recovery rate of the exogenous added BDNF in the measured serum samples was more than 95%. All measurements were performed in triplicate, and the average value was used as the measured value.

### 2.4. Statistical Analysis

All statistical analyses were performed using EZR 1.60 and STATA 17.0. *p*-values were calculated using Welch’s *t*-test for the difference in BDNF levels between the two groups at 0, 2, 4, and 8 weeks. Adjusted *p*-values were calculated using a multiple regression analysis to adjust for age and sex.

Using a mixed-effects model, changes in BDNF levels at 2, 4, and 8 weeks were compared to baseline values (0 week) in the response and non-response groups, respectively. Interactions between responders and non-responders were calculated using a mixed-effects model. Interactions were evaluated for the difference in the slope of BDNF values between the two groups compared to baseline values (0 week) at 2, 4, and 8 weeks. 

Three mixed-effects models were created: a response group, a non-response group, and both groups combined for the interaction, with age and sex substituted as covariates. The intercept was set to random effects and time (i.e., 0, 2, 4, and 8 weeks) was treated as a nominal variable. In the mixed effects model, an ANOVA (global test) was conducted to confirm the null hypothesis that all means and changes are equal at the four time points. The validity of the mixed effects model was confirmed by using histograms of the normality of the residuals. All tests were two-tailed, and a *p*-value < 0.05 was considered statistically significant. Demographic data were expressed as mean (standard deviation) or percentage, and statistical values obtained were expressed as mean (standard error).

### 2.5. Ethical Statement

The study protocol was approved by the Ethics Committee of the University of Occupational and Environmental Health (Approved date: August 2018, Approved#: UOEHCRB21-057). Written informed consent was obtained from all participants.

## 3. Results

### Patient Demographics and BDNF Levels

There were no significant differences in sex, age, and HAMD scores before drug administration between the responder and non-responder groups (Table 1). Serum BDNF levels at 0, 2, 4, and 8 weeks were not significantly different between the responder and non-responder groups (Table 3). The responder group showed statistically significant changes in serum BDNF 0 (baseline) and serum BDNF 8. In contrast, the non-responder group showed no statistically significant changes from baseline (Table 4 and Table 5, Figure 2 and Figure 3). There was a statistically significant interaction between serum BDNF 0 (baseline) and serum BDNF 8 between the responder and non-responder groups (Table 4).

## 4. Discussion

The results of the current study showed that serum BDNF concentrations increased significantly after 8 weeks in the paroxetine, escitalopram, and duloxetine response groups, but not after 2 or 4 weeks. However, serum BDNF concentrations did not increase at any point in time in the non-response group. We also found that serum BDNF levels at week 0 (baseline) did not statistically differ between the responder group and the non-responder group. 

Previous studies have reported that serum BDNF concentrations increased during antidepressant treatment [33,34,47,48]. However, the duration of the response and the type of drug used varied. In particular, selective serotonin receptor inhibitors (SSRIs) and serotonin-norepinephrine receptor inhibitors (SNRIs) have been shown to increase serum BDNF levels after 8 weeks of treatment [34,47,49]. The current study examined serum BDNF concentrations after 8 weeks of paroxetine or milnacipran (an SNRI) treatment and found that serum BDNF concentrations after 8 weeks significantly increased in the group of patients who had responded to paroxetine or milnacipran treatment, while serum BDNF concentrations after 4 weeks were unchanged with treatment with both antidepressants. In contrast, there was no change in serum BDNF concentrations in the group of non-responders. In the second set of 35 untreated patients, the serum BDNF concentration after 8 weeks of treatment with paroxetine, duloxetine, or escitalopram was significantly increased, while the serum BDNF concentration after 2 or 4 weeks of treatment was unchanged in the response group. Taken together, 8 weeks may be necessary for serotonin reuptake inhibitors to increase serum BDNF in the responder group. Our current study demonstrated that mirtazapine, a different class of SSRI or SNRI, could increase serum BDNF at week 4 in the responder group [50]. Thus, mirtazapine might have a rapid effect on the increase in serum BDNF. In a study of sertraline, venlafaxine, and escitalopram, elevation in the serum BDNF level was observed at 5 weeks and 6 months post-dose in the sertraline group and at 6 months post-dose in the venlafaxine group, whereas no change was observed in the non-responder group at any point in time. In contrast, the escitalopram group showed no increase in serum BDNF after 6 months [51]. Taking these findings into account, the changes in serum or plasma BDNF seem to be inconsistent. The one important reason for this inconsistence might be a heterogeneity of MD.

A decrease in serum BDNF levels in the early phase of SSRI treatment may be associated with a later SSRI response in adolescents with MD [39]. A previous report demonstrated that responders to treatment (≥50% improvement in depression ratings) had higher pre-treatment BDNF levels than did non-responders [52,53]. On the other hand, we found that baseline serum BDNF did not differ between the groups. A possible explanation is that baseline serum BDNF levels vary between patients (Figure 4a,b). If we confirm the result using a larger sample, serum BDNF levels the in non-responder group might be higher than in the responder group. In short, this might be a type II error. A study reported that plasma BDNF was not significantly changed after 1–2 days of single ketamine administration compared to placebo, which does not support the hypothesis that ketamine treatment increases BDNF plasma levels in patients with MD [54]. Another report demonstrated that BDNF was significantly elevated for only 1 week following the first ketamine infusion in those classified as responders [43]. No correlations were found between plasma BDNF levels and response to venlafaxine and paroxetine treatment at week 10 in patients with MD [55]. Treatment with venlafaxine for 4 weeks decreased serum BDNF levels, whereas treatment with mirtazapine for 4 weeks increased serum BDNF levels in patients with MD [56]. Treatment with mirtazapine for 12 weeks increased serum BDNF levels, which is associated with its response [57]. Based on these findings, the relationship between antidepressants, duration of treatment, and treatment response is also inconsistent and complicated. 

We previously reported that early changes in serum BDNF levels (from week 0 and week 4) did not predict the response to treatment with SSRIs [58]. A recent systematic review and network meta-analysis found a significant effect of antidepressants on increased BDNF levels [standardized mean difference (SMD) = 0.62; 95% confidence interval (CI) = 0.31–0.94, Z = 3.92, *p* < 0.0001] [39]. Increases in BDNF levels over time were also associated with significant decreases in HAMD scores (SMD = 2.78, 95% CI = 2.31–3.26, Z = 11.57, *p* < 0.00001). The review also reported that SNRIs showed higher effect sizes than SSRIs (0.92 vs. 0.68). In addition, four antidepressants were analyzed separately for their role in increasing BDNF levels. Among these, only sertraline showed a significant increase in BDNF levels after treatment (SMD = 0.53, 95% CI = 0.13–0.93, Z = 2.62, *p* = 0.009), while venlafaxine, paroxetine, and escitalopram did not. In any case, a follow-up study with a large sample and a uniform protocol is needed regarding antidepressant treatment and changes in serum or plasma BDNF in MD.

Furthermore, it has been reported that electroconvulsive therapy (ECT) also could alter serum BDNF levels in MD patients [59,60,61,62,63], but other reports did not produce the same findings [64,65]. Repetitive transcranial magnetic stimulation (rTMS) also increases serum BDNF levels [66,67,68,69,70]. Taken together, the BDNF pathway is a common pathway for antidepressants, ECT, and rTMS to improve depressive symptoms, which is no longer controversial.

In the current study, antidepressants generally had a significant effect on the increase in serum BDNF levels after 8 weeks. Our prospective study of serum BDNF concentrations in a relatively small number of patients demonstrated that antidepressant-responsive patients had the first significant increase after 8 weeks of treatment, while non-responders showed no change at any point in time. These results are consistent with those of Zhou et al. [39] and did not contradict our previous report demonstrating that early changes in serum BDNF levels (from week 0 and week 4) did not predict the response to treatment with SSRIs [57]. Another systematic review and meta-analysis reported that peripheral measurements of BDNF are inadequate predictors of treatment response in treatment-refractory MD patients [69]. In our previous reports [71,72,73], catecholamine metabolites were altered after 4 weeks in the antidepressant response group, whereas BDNF was altered after 8 weeks in the antidepressant response group in the present results. These results suggest that changes in blood catecholamine metabolites precede changes in blood BDNF. Based on these findings, serum BDNF may be a candidate as a predictive factor for treatment response; however, it is difficult to predict the treatment of MD simply from BDNF changes alone. Combining BDNF changes with other biomarkers including microRNA [74], DNA methylation [75], proteomic markers [76], genetic information [77], and imaging findings [78] may help to more accurately predict treatment response and prognosis.

This study has several important limitations. The first is the flexible dose design, with the type and dosage of antidepressants left to the discretion of the attending physician. Second, the number of patients was small (n = 35), and we did not perform a power analysis to determine the sample number. In addition, basal levels vary between patients, and the levels in the same patients at different points in time is much more consistent within the same patient (see Table 4), which might lead to the result that baseline serum BDNF levels did not differ in both groups. If we collected larger samples of serum BDNF levels in the non-responders, they might have been higher than in the responder group. Alternatively, the trajectory of serum BDNF changes might be important, but not the basal levels. In any case, further studies are required to explain this point. Third, there was no placebo group. Another limitation is the short clinical course of the patients, who were followed up for only 8 weeks after antidepressant administration. We are now undertaking a large-scale study including a placebo group, with a longer duration follow-up study (considering the above problems) to overcome these limitations. We hope to eventually establish a system that more accurately predicts antidepressant responsiveness in MD patients using not only serum BDNF data, but also technologies such as genomics, epigenomics, transcriptomics, proteomics, metabolomics, and connectomes [79].

## 5. Conclusions

In patients with first-episode and drug-naive MD treated with antidepressants, serum BDNF concentrations in the treatment response group increased significantly only after 8 weeks but not after 2 or 4 weeks of treatment. In contrast, no change in the serum BDNF concentration was observed in the non-responder group at any point in time. The difference of the changes of serum BDNF levels between the responders and the non-responders to antidepressants might be complicated, and must be further elucidated for each antidepressant. We believe that combined serum BDNF changes with other information including genomics, epigenomics, transcriptomics, proteomics, metabolomics, and connectomes could become an accurate method of prediction for drug response in MD patients. We urgently need to establish such a system for clinical practice. 

## Figures and Tables

**Figure 1 biomedicines-11-00584-f001:**
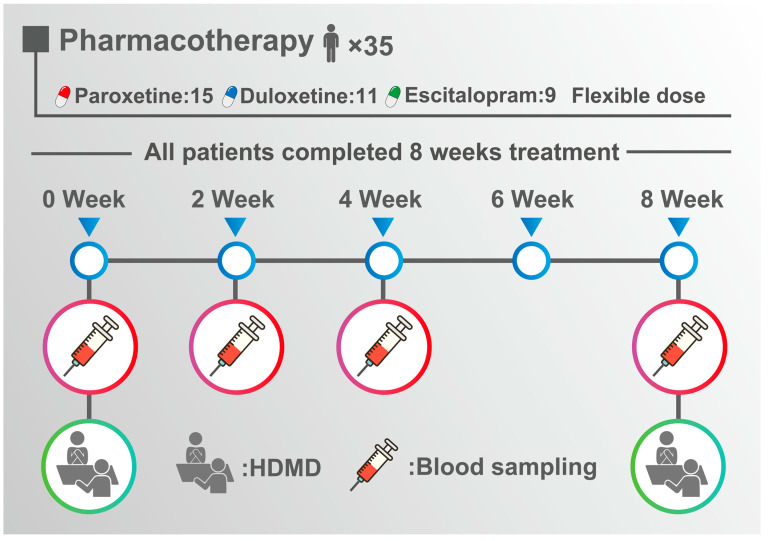
Study protocol.

**Figure 2 biomedicines-11-00584-f002:**
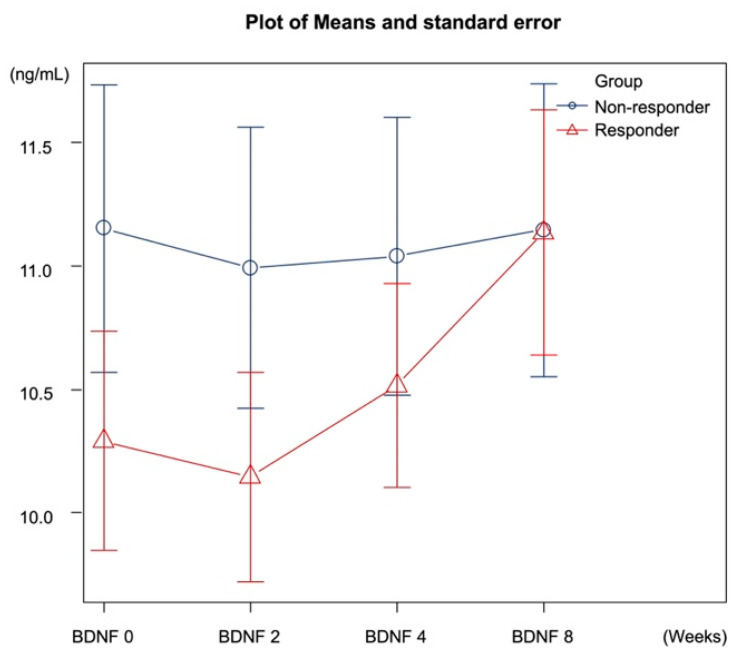
Changes in serum BDNF levels in the responders and non-responders. Data was expressed as mean and standard error.

**Figure 3 biomedicines-11-00584-f003:**
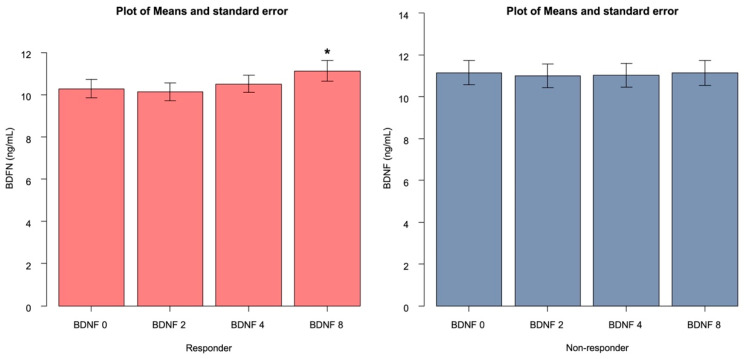
Changes in serum BDNF levels in the responders (red column) and non-responders (blue column). Data was expressed as mean and standard error. ANOVA (global test’s *p*-value for interaction analysis in mixed effect model; * *p* = 0.0025.

**Figure 4 biomedicines-11-00584-f004:**
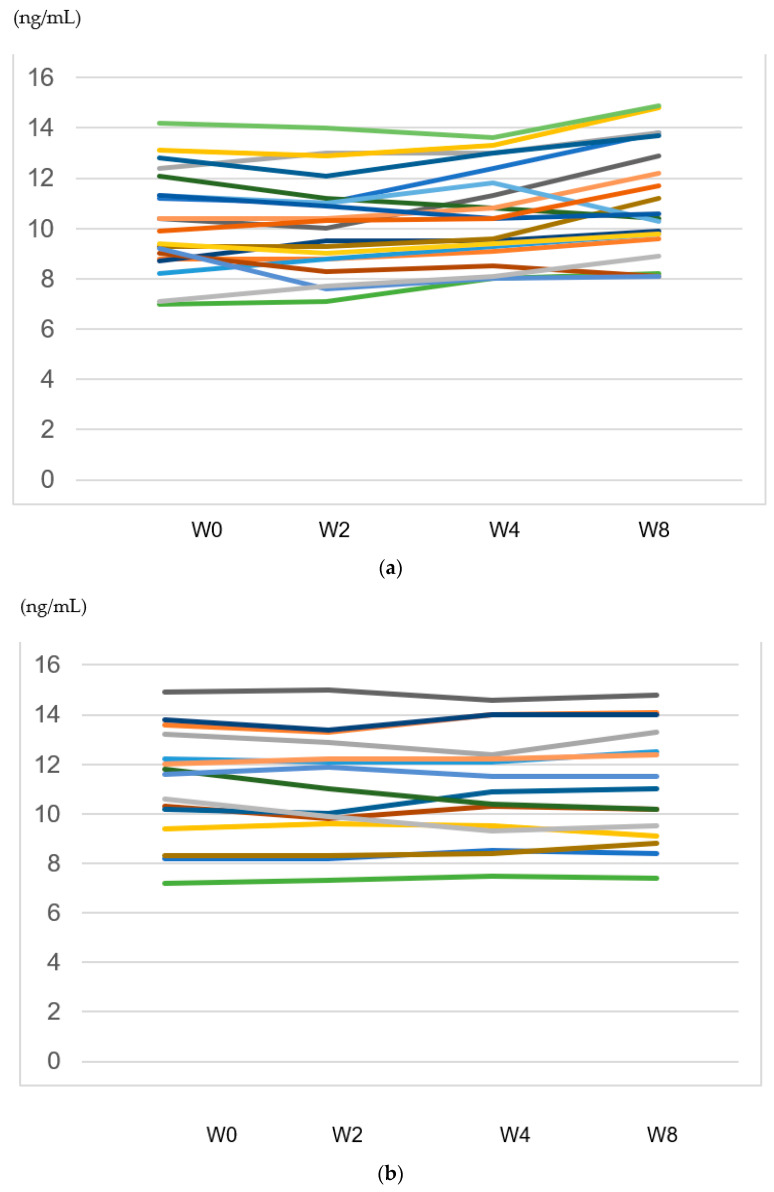
(**a**) Trajectory of serum BDNF levels in each patient with responders. (**b**) Trajectory of serum BDNF levels in each patient with responders.

**Table 1 biomedicines-11-00584-t001:** Background of treatment response group and treatment cost response group.

	Responder (n = 20)	Non-Responder (n = 15)
Age, year	37.8 (7.21)	35.3 (6.67)
Sex (male), (%)	8 (40.0%)	7 (46.6%)
Smokers, (%)	4 (20%)	4 (26%)
History of abuse, (%)	0 (0%)	0 (0%)
Comorbidity, (%)	0 (0%)	0 (0%)
Duration of illness, weeks	8.41 (2.31)	8.80 (2.47)
HAMD 0 (points)	23.3 (2.87)	23.0 (2.42)
HAMD 8 (points)	9.45 (2.03)	15.4 (2.26)

Data are expressed as mean (standard deviation).

**Table 2 biomedicines-11-00584-t002:** Drugs and daily dose of the patients.

Drug	#	Min. Dose (mg/day)	Max. Dose (mg/day)	Mean (SD) (mg/day)
Escitalopram	9	10	20	18.8 (3.1)
Paroxetine	11	20	40	49.0 (9.9)
Duloxetine	15	40	60	32.0 (9.7)

**Table 3 biomedicines-11-00584-t003:** Serum BDNF concentrations in treatment responders and non-responders.

	Responder	Non-Responder	*p*-Value	Adjusted *p*-Value
BDNF 0	10.2 (1.98)	11.1(2.25)	0.24	0.24
BDNF 2	10.1 (1.90)	10.9 (2.21)	0.24	0.24
BDNF 4	10.5 (1.84)	11.0 (2.18)	0.45	0.52
BDNF 8	11.1 (2.22)	11.1 (2.30)	0.98	0.97

Data are expressed as mean (standard deviation). The *p*-value was calculated by Welch’s *t*-test. Adjusted *p*-values were calculated using multiple regression analysis adjusted for age and sex.

**Table 4 biomedicines-11-00584-t004:** Changes in serum BDNF (Change from BDNF 0).

	Estimate	95% CI	Standard Error	t-Value	Adjusted *p*-Value
*Responder*					
BDNF 2	−0.145	−0.520, 0.230	0.193	−0.750	0.45
BDNF 4	0.225	−0.150, 0.600	0.193	1.164	0.24
BDNF 8	0.845	0.469, 1.220	0.193	4.371	<0.001 *
*Non-responder*					
BDNF 2	−0.160	−0.406, 0.086	0.127	−1.255	0.21
BDNF 4	−0.113	−0.360, −0.133	0.127	−0.889	0.37
BDNF 8	−0.006	−0.253, 0.240	0.127	−0.052	0.95

ANOVA (global test’s *p*-value for responder group in mixed effect model < 0.001 *). ANOVA (global test’s *p*-value for non-responder group in mixed effect model = 0.51). Adjusted *p*-values were calculated using a mixed effects model adjusted for age and sex. CI means confidence interval.

**Table 5 biomedicines-11-00584-t005:** Interaction analysis between the two groups (Change from BDNF 0).

	Estimate	95% CI	Standard Error	t-Value	Adjusted *p*-Value
Group BDNF 2	0.015	−0.464, 0.494	0.249	0.060	0.95
Group BDNF 4	0.338	−0.140, 0.817	0.249	1.356	0.17
Group BDNF 8	0.851	0.372, 1.330	0.249	3.413	<0.001 *

ANOVA (global test’s *p*-value for interaction analysis in mixed effect model = 0.0025 *). The adjusted *p*-value was calculated by a mixed effects model adjusted for age and sex. CI means confidence interval.

## Data Availability

The data presented in this study is available upon request from the corresponding author.

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
