# Peer review of "The Serum Brain-Derived Neurotrophic Factor Increases in Serotonin Reuptake Inhibitor Responders Patients with First-Episode, Drug-Naïve Major Depression"

_biomedicines, 2023, doi:10.3390/biomedicines11020584_

Round 1

Reviewer 1 Report

The manuscript “Changes in serum brain-derived neurotrophic factor in patients with first-episode and untreated major depression” describes small scale but nonetheless interesting clinical study. The study was well designed, the methodology is sound and well described. The conclusions are well balanced and limitations are clearly discussed. However the study could still be slightly improved:

1.       The authors do not discuss the fact that BDNF lewels in non-reposnder gourp at 0 weeks (baseline) is actually on the level of responders group at same time (albeit not significant statistically), which would probably need much larger studies. This might arise from high statistical variation in baseline levels, which was also seen in other studies. And statistic suggest that while basal levels vary between patients, the level in same patients at different time point is  much more consistent within same patient (Table 3). But alternative explanation that Non-responders might have basally higher levels of BDNF should be discussed (or the fact that not basal level but trajectory of changes matter).

2.       Linked with previous point It would be also beneficial if the authors could provide graph with individually ploted trajectories of change in BDNF so they can be more easily assessed by the reader i.e. data in table 3 with individual data points plotted on graph. This could be separated for clarity for reponders/nonerepsonders and which subgraph for each of tested drugs drugs.

3.       Please provide more details (e.g. in tabular form) of exact dosing regimens of patients (e.g. dose + number of patients on specified dose).

Author Response

To reviewers

We responded your comments as follows.

Dear Authors,

The brain-derived neurotrophic factor (BDNF) involved in the pathogenesis of major depressive disorder (MDD) is considered to mediate the action of antidepressants. In this manuscript, entitled ‘Changes in serum brain-derived neurotrophic factor in patients with first-episode and untreated major depression’, Yoshimura and colleagues studied the serum BDNF levels in antidepressant treatment responders and treatment non-responders of the first-episode, drug-naïve patients.
The main strength of this manuscript is that it addresses a timely and interesting topic, providing innovative research presenting that treatment responders showed significant increase in BDNF levels, while treatment non-responders did not.
In general, I think the idea of this review article is interesting and the authors’ fascinating observations on this timely topic may be of interest to the readership of Biomedicines. However, some comments as well as some crucial evidence should be included to support the authors’ argumentation to improve its adequacy, its readability, and thus the quality of the manuscript, prior to the publication. My overall opinion is to publish this research article after the authors have carefully considered reviewers’ comments and my suggestions below during the peer-review session.
Please consider the following comments:
1. Please present the title concise, self-explanatory and stating the most significant results of this study.

Our reply:

We changed the title as follows.

Trajectory of serum BDNF in first-episode, drug-naïve patients with major depression

  1. Abstract: Please abridge the abstract to 200 words according to the guidelines of the journal (https://www.mdpi.com/journal/biomedicines/instructions) and proportionally present the subsection in the following order without subheadings: the background, the objectives, the methods, the results, and the conclusion. The background should include the general background (one to two sentences), the specific background (two to three sentences), and current issue addressed to this study (one sentence). The result should include one sentence describing the main result using such words like “Here we show”. The conclusion should write the potential and the advance this study has provided in the field and finally a broader perspective (two to three sentences) readily comprehensible to a scientist in any discipline.

Our reply:

We changed abstract as follows within 200 words.

The trajectory of serum BDNF remains obscure. The present study aimed to compare the changes in serum BDNF concentrations in first-episode, drug-naive patients with MD treated with antidepressants between treatment-response and treatment-nonresponse groups. The study included 35 inpatients and outpatients composed of 15 males and 20 females aged 36.7 ± 6.8 years at the Department of Psychiatry of our University Hospital. All patients met the DSM-5 diagnostic criteria for MD and were first-episode drug-naive. The antidepressants administered included paroxetine, duloxetine, and escitalopram. Severity of depressive state was assessed using the 17-item HAMD before and 8 weeks after drug administration. Responders were defined as those whose total HAMD scores at 8 weeks had decreased by 50% or more compared to those before drug administration, while non-responders were those whose total HAMD scores had decreased by less than 50%. Serum BDNF concentration was measured using ELISA. Here we showed that serum BDNF levels were not significantly different at any points between the two groups. The responder group, but not the non-responder group showed statistically significant changes in serum BDNF 0 and serum BDNF 8. The difference in trajectory of BDNF might reflect the clinical response to pharmacotherapy in MD patients. (198 words)

  1. Keywords: Please list ten keywords and use as many as possible in the title and the first two sentences of the abstract.

Our reply:

We picked up 10 keywords as follows.

brain-derived neurotrophic factor; serum; trajectory; major depression; first-episode; drug-naïve; Diagnostic and Statistical Manual of Mental Disorders Fifth Edition; Hamilton Rating Scale for Depression; responders; non-responders

  1. A graphical abstract is highly recommended.

Our reply:

We made a graphical abstract.

  1. Introduction: The authors need to fully expand this section, as it is short of information on the main constructs of this study, which should be understood to a reader in any discipline and make persuasive enough to put forward the main purpose of current research the authors have conducted and the specific purpose the authors have intended by this study. I would like to encourage the authors to present the introduction starting with the general background, proceeding to the specific background, and finally the current issue addressed to this study, leading to the objectives. Those main structures should be organized in a logical and cohesive manner.

Our reply:

We expanded the Introduction as follows according to your suggestion.

Brain-derived neurotrophic factor (BDNF) is often implicated in the pathophysiology of major depression (MD). In particular, BDNF plays a major role in neuronal growth and survival, functions as a neurotransmitter modulator, and contributes to neuronal plasticity. BDNF stimulates and regulates the growth of new neurons from neural stem cells (i.e., neurogenesis), BDNF has been linked to synaptic re-modeling, being able to both induce and be induced by long-term potentiation (1,2). Both BDNF protein and mRNA have been detected in various brain regions, including the olfactory bulb, cerebral cortex, hippocampus, basal forebrain, mesencephalon, hypothalamus, brainstem, and spinal cord (3-8). Therefore, it has been suggested that abnormalities in BDNF in the brain are associated with the pathogenesis of MD (9-15). Furthermore, it has been reported that BDNF gene Val66Met polymorphism is associated with vulnerability to MD (16,17). Several meta-analyses have shown that serum and plasma levels of BDNF were significantly decreased in patients with MD compared to healthy controls (18-23). Similarly, in our previous work, we also reported that serum BDNF concentrations were significantly lower in untreated patients with MD than in healthy controls (22,23). A meta-analysis showed that various antidepressant treatments increase serum and plasma BDNF concentrations in patients with MD (22,26,27). We also previously found that paroxetine and milnacipran significantly increased serum BDNF concentrations after 8 weeks in untreated patients with MD in the treatment response group (24,25). Although there have been reports of increased serum and plasma BDNF levels after antidepressant treatment in patients with MD (10,11), the findings of the relationship between changes in serum/plasma BDNF levels and response to pharmacotherapy, however controversial, cannot be fully elucidated. Response to duloxetine was associated with a higher baseline serum BDNF level and greater reduction of the Hamilton Rating Scale for Depression (HAMD) scores for major depression (28). The absence of an early increase of serum BDNF levels in conjunction with early non-response to antidepressants can be a highly specific peripheral biomarker predictive for treatment failure of pharmacotherapy in MD (29). Alternatively, the combination of an early increase (day 7) of plasma BDNF with early reduction of the HAMD scores could be a useful predictive marker for pharmaco-treatment in MD (30). A decrease of serum BDNF levels at week 2 of selective serotonin reuptake inhibitor (SSRI) treatment might be associated with later SSRI response in adolescents with MD (31). Pretreatment serum BDNF levels have been reported to be correlated with antidepressant responses, and responders to treatment improvement in severity of MD had higher pretreatment serum BDNF levels than did non-responders (32). In short, the results of the time course of serum/plasma BDNF levels and response to antidepressants were not consistent and remain obscure. Moreover, there are no reports of detailed observations of the time course of serum/plasma BDNF levels and the response to antidepressants in MD patients. Thus, this study aimed to compare the trajectory of serum BDNF concentrations between the treatment-response and treatment-nonresponse groups in first-episode, drug-naive patients with MD treated with antidepressants.

  1. Materials and Methods: I recommend presenting a figure clarifying the schedule of this study and I suggest citing more references to ensure the integrity and the reliability in evidence that the authors built the study design and applied the methodology conducted in this study. Also, it deserves to present more demographical information.  Please clarify how the authors determined the sample numbers. Did they conduct a power analysis?

Our reply:

We inserted the additional demographic data in Table 1.  We made a figure clarifying schedule of the present study as Figure 1. We described the assay of serum BDNF in detail as follows.

In brief, 96-well microplates were coated with anti-BDNF monoclonal antibody and incubated at 4℃ for 18 hours. The plates were incubated in a blocking buffer for 1 hour at room temperature. The samples diluted with assay buffer by 100-times and BDNF standards were kept at room temperature under conditions of horizontal shaking for 2 hours, followed by washing with the appropriate washing buffer. The plates were incubated with antihuman BDNF polyclonal antibody at room temperature for 2 hours and washed with the washing buffer. The plates were then incubated with anti-IgY antibody conjugated to horseradish peroxidase for 1 hour at room temperature, and incubated in peroxidase substrate and tetramethylbenzidine solution to induce a color reaction. The reaction was stopped with 1 mol/L hydrochloric acid. The absorbance at 450 nm was measured with an Emax automated microplate reader. Measurements were performed in duplicates. The standard curve was linear from 0.5 ng/mL to 50ng/mL, and the detection limit was 5 ng/mL. Cross-reactivity to related neurotrophins (NT-3, NT-4, NGF) was less than 3%. Intra- and interassay coefficients of variation were about 5% and 7%, respectively. The recovery rate of the exogenous added BDNF in the measured serum samples was more than 95%. All measurements were performed in triplicate, and the average value was used as the measured value.

Our reply:
We did not perform the power analysis to determine the sample number. Thus, we added the point in the limitation as follows.

This study has important limitations. First is the flexible dose design, with the type and dosage of antidepressants left to the discretion of the attending physician. Second, the number of patients was small (n = 35), and additionally we did not perform the power analysis to determine the sample number. Third, there was no placebo group. Another limitation is the short clinical course of the patients, who were followed up for only 8 weeks after antidepressant administration. A large-scale study including a placebo group and longer duration follow up should be conducted in the future to overcome these limitations.

  1. Results:  Please present all figures in color and all statistical values in tabular form.

Our reply:

We remade the color figures, and all statistical values in tabular form.

  1. Discussion: The authors need to totally reorganize and fully expand this section. Starting with the summary of the previous section (Results), the authors need to develop discussion on the potential of this study complementing as the extension of the previous work, the implication of the findings of this study, how this study could facilitate future research, the ultimate goal, the challenge, the knowledge and the technology necessary to achieve this goal, the statement about this field in general, and finally the importance of this line of research. It is particularly important to present its limit and its merit, and its potential translation of this study to clinical practice.

Our reply:

We extended and arranged the Discussion as follows.

The results of the current study showed that serum BDNF concentrations increased significantly after 8 weeks in the paroxetine, escitalopram, and duloxetine response groups, but not after 2 or 4 weeks. However, serum BDNF concentrations did not increase at any time point in the non-response group. Previous studies have reported that serum BDNF concentrations increased during antidepressant treatment (19,20,33,34). However, the duration of the response and the type of drug used varied. In particular, selective serotonin receptor inhibitors (SSRIs) and serotonin-norepinephrine receptor inhibitors (SNRIs) have been shown to increase serum BDNF levels after 8 weeks of treatment (22,35). The current study examined serum BDNF concentrations after 8 weeks of paroxetine treatment and found that serum BDNF concentrations after 8 weeks significantly increased in the group of patients who had responded to paroxetine treatment, while serum BDNF concentrations after 4 weeks were unchanged. In contrast, there was no change in serum BDNF concentration in the group of non-responders. In the second set of 35 untreated patients, the serum BDNF concentration after 8 weeks of treatment with paroxetine, duloxetine, or escitalopram was significantly increased, while the serum BDNF concentration after 2 or 4 weeks of treatment was unchanged in the response group. In a study of sertraline, venlafaxine, and escitalopram, elevation in serum BDNF level was observed at 5 weeks and 6 months post-dose in the sertraline group and at 6 months post-dose in the venlafaxine group, whereas no change was observed in the non-responder group at any time point. In contrast, the escitalopram group showed no increase in serum BDNF after 6 months (36).

A decrease in serum BDNF levels in the early phase of SSRI treatment may be associated with a later SSRI response in adolescents with MD (37). A study reported that plasma BDNF was not significantly changed after 1-2 days of single ketamine administration compared to placebo, which does not support the hypothesis that ketamine treatment increases BDNF plasma levels in patients with MD (38). Another report demonstrated that BDNF was significantly elevated only 1 week following the first ketamine infusion in those classified as responders (39). No correlations were found between plasma BDNF levels and response to venlafaxine and paroxetine treatment at week 10 in patients with MD (40). Treatment with venlafaxine for 4 weeks decreased serum BDNF levels, whereas treatment with mirtazapine for 4 weeks increased serum BDNF levels in patients with MD (41). Treatment with mirtazapine for 12 weeks increased serum BDNF levels, which is associated with its response (42). Based on these findings, the relationship between antidepressants, duration of treatment, and treatment response is inconsistent and complicated.

We previously reported that early changes in serum BDNF levels (from week 0 and week 4) did not predict the response to treatment with SSRIs (43). A recent systematic review and network meta-analysis found a significant effect of antidepressants on increased BDNF levels [standardized mean difference (SMD) = 0.62; 95% confidence interval (CI) = 0.31-0.94, Z = 3.92, p < 0.0001] (25). Increases in BDNF levels over time were also associated with significant decreases in HAMD scores (SMD = 2.78, 95% CI = 2.31-3.26, Z = 11.57, p < 0.00001). The review also reported that SNRIs showed higher effect sizes than SSRIs (0.92 vs. 0.68). In addition, four antidepressants were analyzed separately for their role in increasing BDNF levels. Among these, only sertraline showed a significant increase in BDNF levels after treatment (SMD = 0.53, 95% CI = 0.13-0.93, Z = 2.62, p = 0.009), while venlafaxine, paroxetine, and escitalopram did not.

Furthermore, it has been reported that electroconvulsive therapy (ECT) also could alter serum BDNF levels in MD patients (44-48), but other reports did not produce the same findings (49,50). Repetitive transcranial magnetic stimulation (rTMS) also increases serum BDNF levels (51-54). Taken together, BNNP pathway is common pathway for antidepressants, ECT, and rTMS improve depressive symptoms, which is not still controversial.

In the current study, antidepressants generally had a significant effect on the increase in serum BDNF levels after 8 weeks. Our prospective study of serum BDNF concentrations in a relatively small number of patients demonstrated that antidepressant-responsive patients had the first significant increase after 8 weeks of treatment, while non-responders showed no change at any time point. These results are consistent with those of Zhou et al. (25) and did not contradict our previous report demonstrating that early changes in serum BDNF levels (from week 0 and week 4) did not predict the response to treatment with SSRIs (43). Another systematic review and meta-analysis reported that peripheral measurements of BDNF are inadequate predictors of treatment response in treatment-refractory MD patients (55). In our previous reports (56-58), catecholamine metabolites were altered after 4 weeks in the antidepressant response group, whereas BDNF was altered after 8 weeks in the antidepressant response group in the present results. These results suggest that changes in blood catecholamine metabolites precede changes in blood BDNF. Based on these findings, serum BDNF may be a candidate as a predictive factor for treatment response; however, it is difficult to predict treatment of MD simply from BDNF trajectory alone. Combining BDNF trajectory with other biomarkers and imaging findings may help to more accurately predict treatment response and prognosis.

This study has important limitations. First is the flexible dose design, with the type and dosage of antidepressants left to the discretion of the attending physician. Second, the number of patients was small (n = 35), and additionally we did not perform power analysis to determine the sample number. Third, there was no placebo group. Another limitation is the short clinical course of the patients, who were followed up for only 8 weeks after antidepressant administration. A large-scale study including a placebo group and longer duration follow up should be conducted in the future to overcome these limitations.

  1. Conclusion: I think that this section would benefit from a paragraph presenting some thoughtful as well as in-depth considerations by the authors as experts to convey the take-home message, as it is very descriptive but not enough theoretical as a conclusion should be. The authors should make their effort to explain the theoretical implication as well as the translational application of their research.

Our reply

We extended the Conclusion section as follows.

In patients with first-episode and drug-naive MD treated with antidepressants, serum BDNF concentrations in the treatment response group increased significantly only after 8 weeks but not after 2 or 4 weeks of treatment. In contrast, no change in the serum BDNF concentration was observed in the non-responder group at any time point.

The difference of the trajectory of serum BDNF levels between the responders and non-responders to antidepressants might be complicated, and must be further elucidated for each antidepressant, and we must also follow the trajectory until at remission of MD.

In any case, serum or plasma BDNF levels could not be a robust biomarker for the prediction of antidepressants in MD patients at present.

  1. References: I suggest presenting more references to support the authors’ argumentation needed to be addressed to improve the quality, its adequacy, and its readability of the manuscript. Typically, a paper like this cites 60-70 references.
    Overall, the manuscript contains one figure, four tables, and 25 references. I believe that the manuscript may carry important value in showing the difference in BDNF levels between treatment responders and treatment non-responders of MDD patients. I hope that, after careful revisions, the manuscript can meet the Journal’s high standards for publication.
    I declare no conflict of interest regarding this manuscript.

Our reply:

We inserted additional references, and total reference number is 60.

The revised manuscript is suitable for publication in Biomedicine.

Best regards,

Reiji

Reviewer 2 Report

Yoshimura and colleagues in this research article entitled ‘Changes in serum brain-derived neurotrophic factor in patients with first-episode and untreated major depression’, investigated the changes in serum brain-derived neurotrophic factor (BDNF) concentrations in first-episode, drug-naive patients with major depression treated with antidepressants, comparing behavioral responses and serum concentrations between treatment-response group and treatment-nonresponse group. 

In general, I think the idea of this article is really interesting and the authors’ fascinating observations on this timely topic may be of interest to the readers of Biomedicines. However, some comments, as well as some crucial evidence that should be included to support the authors' argumentation, needed to be addressed to improve the quality of the manuscript, its adequacy, and its readability prior to the publication in the present form. My overall judgment is to publish this paper after the authors have carefully considered my suggestions below, in particular reshaping parts of the ‘Introduction’ and ‘Methods’ sections by adding more evidence.

Please consider the following comments:

Abstract: According to the Journal’s guidelines, the abstract should be presented as a single paragraph without sub-headings. Also, it should be a total of about 200 words maximum. Please correct the actual one.

Keywords: I would suggest adding ‘Hamilton Depression Rating Scale (HAMD)’ as a keyword.

Introduction: The ‘Introduction’ section is well-written and nicely presented, with a good balance of descriptive text and information about pathomechanisms and neurobiological alterations underlying major depressive disorder. Nevertheless, I believe that more information about pathophysiology and core features of this disorder will provide a better and more accurate background, because as it stands, this information is not highlighted in the text. I suggest to begin with a theoretical explanation of depressive disorder and the role of specific brain areas, like prefrontal cortex, in the pathophysiology of this disorder. In this regard, I would suggest to add more information on pathological neural substrates of depression disorder, for example focusing on ‘Dissecting Neurological and Neuropsychiatric Diseases: Neurodegeneration and Neuroprotection’ and on structural as well as functional abnormalities of prefrontal cortex that may affect patients’ cognitive impairments (https://doi.org/10.3390/biomedicines10123189). In my opinion, authors could further explore relationship between the molecular regulation of higher-order neural circuits and neuropathological alterations in this neuropsychiatric disorder (https://doi.org/10.3390/biomedicines10081897), in order to provide more insights on antidepressant associated with upregulation of serum BDNF in depressed patients.

Results: I suggest rewriting this section more accurately. To properly present experimental findings, I think that authors should provide full statistical details (like degree of freedom or post-hoc utilized) also in the main text, to ensure in-depth understanding and replicability of the findings.

In my opinion, I think the ‘Conclusions’ paragraph would benefit from some thoughtful as well as in-depth considerations by the authors, because as it stands, it is very descriptive but not enough theoretical as a discussion should be. Authors should make an effort, trying to explain the theoretical implication as well as the translational application of their research.

In according to the previous comment, I would ask the authors to include a proper and defined ‘Limitations and future directions’ section before the end of the manuscript, in which authors can describe in detail and report all the technical issues brought to the surface.

References: Authors should consider revising the bibliography, as there are several incorrect citations. Indeed, according to the Journal’s guidelines, they should provide the abbreviated journal name in italics, the year of publication in bold, the volume number in italics for all the references. Also, please correct in-text citations: reference should be numbered, and placed in square brackets [ ] (for example [1]).

I hope that, after these careful revisions, this paper can meet the Journal’s high standards for publication. 

I am available for a new round of revision of this article. 

Best regards,

Reviewer 

Author Response

To reviewer

We revised the manuscript according to your comments as follows.

The brain-derived neurotrophic factor (BDNF) involved in the pathogenesis of major depressive disorder (MDD) is considered to mediate the action of antidepressants. In this manuscript, entitled ‘Changes in serum brain-derived neurotrophic factor in patients with first-episode and untreated major depression’, Yoshimura and colleagues studied the serum BDNF levels in antidepressant treatment responders and treatment non-responders of the first-episode, drug-naïve patients.
The main strength of this manuscript is that it addresses a timely and interesting topic, providing innovative research presenting that treatment responders showed significant increase in BDNF levels, while treatment non-responders did not.
In general, I think the idea of this review article is interesting and the authors’ fascinating observations on this timely topic may be of interest to the readership of Biomedicines. However, some comments as well as some crucial evidence should be included to support the authors’ argumentation to improve its adequacy, its readability, and thus the quality of the manuscript, prior to the publication. My overall opinion is to publish this research article after the authors have carefully considered reviewers’ comments and my suggestions below during the peer-review session.
Please consider the following comments:
1. Please present the title concise, self-explanatory and stating the most significant results of this study.

Our reply:

We changed the title as follows.

Trajectory of serum BDNF in first-episode, drug-naïve patients with major depression

  1. Abstract: Please abridge the abstract to 200 words according to the guidelines of the journal (https://www.mdpi.com/journal/biomedicines/instructions) and proportionally present the subsection in the following order without subheadings: the background, the objectives, the methods, the results, and the conclusion. The background should include the general background (one to two sentences), the specific background (two to three sentences), and current issue addressed to this study (one sentence). The result should include one sentence describing the main result using such words like “Here we show”. The conclusion should write the potential and the advance this study has provided in the field and finally a broader perspective (two to three sentences) readily comprehensible to a scientist in any discipline.

Our reply:

We changed abstract as follows within 200 words.

The trajectory of serum BDNF remains obscure. The present study aimed to compare the changes in serum BDNF concentrations in first-episode, drug-naive patients with MD treated with antidepressants between treatment-response and treatment-nonresponse groups. The study included 35 inpatients and outpatients composed of 15 males and 20 females aged 36.7 ± 6.8 years at the Department of Psychiatry of our University Hospital. All patients met the DSM-5 diagnostic criteria for MD and were first-episode drug-naive. The antidepressants administered included paroxetine, duloxetine, and escitalopram. Severity of depressive state was assessed using the 17-item HAMD before and 8 weeks after drug administration. Responders were defined as those whose total HAMD scores at 8 weeks had decreased by 50% or more compared to those before drug administration, while non-responders were those whose total HAMD scores had decreased by less than 50%. Serum BDNF concentration was measured using ELISA. Here we showed that serum BDNF levels were not significantly different at any points between the two groups. The responder group, but not the non-responder group showed statistically significant changes in serum BDNF 0 and serum BDNF 8. The difference in trajectory of BDNF might reflect the clinical response to pharmacotherapy in MD patients. (198 words)

  1. Keywords: Please list ten keywords and use as many as possible in the title and the first two sentences of the abstract.

Our reply:

We picked up 10 keywords as follows.

brain-derived neurotrophic factor; serum; trajectory; major depression; first-episode; drug-naïve; Diagnostic and Statistical Manual of Mental Disorders Fifth Edition; Hamilton Rating Scale for Depression; responders; non-responders

  1. A graphical abstract is highly recommended.

Our reply:

We made a graphical abstract.

  1. Introduction: The authors need to fully expand this section, as it is short of information on the main constructs of this study, which should be understood to a reader in any discipline and make persuasive enough to put forward the main purpose of current research the authors have conducted and the specific purpose the authors have intended by this study. I would like to encourage the authors to present the introduction starting with the general background, proceeding to the specific background, and finally the current issue addressed to this study, leading to the objectives. Those main structures should be organized in a logical and cohesive manner.

Our reply:

We expanded the Introduction as follows according to your suggestion.

Brain-derived neurotrophic factor (BDNF) is often implicated in the pathophysiology of major depression (MD). In particular, BDNF plays a major role in neuronal growth and survival, functions as a neurotransmitter modulator, and contributes to neuronal plasticity. BDNF stimulates and regulates the growth of new neurons from neural stem cells (i.e., neurogenesis), BDNF has been linked to synaptic re-modeling, being able to both induce and be induced by long-term potentiation (1,2). Both BDNF protein and mRNA have been detected in various brain regions, including the olfactory bulb, cerebral cortex, hippocampus, basal forebrain, mesencephalon, hypothalamus, brainstem, and spinal cord (3-8). Therefore, it has been suggested that abnormalities in BDNF in the brain are associated with the pathogenesis of MD (9-15). Furthermore, it has been reported that BDNF gene Val66Met polymorphism is associated with vulnerability to MD (16,17). Several meta-analyses have shown that serum and plasma levels of BDNF were significantly decreased in patients with MD compared to healthy controls (18-23). Similarly, in our previous work, we also reported that serum BDNF concentrations were significantly lower in untreated patients with MD than in healthy controls (22,23). A meta-analysis showed that various antidepressant treatments increase serum and plasma BDNF concentrations in patients with MD (22,26,27). We also previously found that paroxetine and milnacipran significantly increased serum BDNF concentrations after 8 weeks in untreated patients with MD in the treatment response group (24,25). Although there have been reports of increased serum and plasma BDNF levels after antidepressant treatment in patients with MD (10,11), the findings of the relationship between changes in serum/plasma BDNF levels and response to pharmacotherapy, however controversial, cannot be fully elucidated. Response to duloxetine was associated with a higher baseline serum BDNF level and greater reduction of the Hamilton Rating Scale for Depression (HAMD) scores for major depression (28). The absence of an early increase of serum BDNF levels in conjunction with early non-response to antidepressants can be a highly specific peripheral biomarker predictive for treatment failure of pharmacotherapy in MD (29). Alternatively, the combination of an early increase (day 7) of plasma BDNF with early reduction of the HAMD scores could be a useful predictive marker for pharmaco-treatment in MD (30). A decrease of serum BDNF levels at week 2 of selective serotonin reuptake inhibitor (SSRI) treatment might be associated with later SSRI response in adolescents with MD (31). Pretreatment serum BDNF levels have been reported to be correlated with antidepressant responses, and responders to treatment improvement in severity of MD had higher pretreatment serum BDNF levels than did non-responders (32). In short, the results of the time course of serum/plasma BDNF levels and response to antidepressants were not consistent and remain obscure. Moreover, there are no reports of detailed observations of the time course of serum/plasma BDNF levels and the response to antidepressants in MD patients. Thus, this study aimed to compare the trajectory of serum BDNF concentrations between the treatment-response and treatment-nonresponse groups in first-episode, drug-naive patients with MD treated with antidepressants.

  1. Materials and Methods: I recommend presenting a figure clarifying the schedule of this study and I suggest citing more references to ensure the integrity and the reliability in evidence that the authors built the study design and applied the methodology conducted in this study. Also, it deserves to present more demographical information.  Please clarify how the authors determined the sample numbers. Did they conduct a power analysis?

Our reply:

We inserted the additional demographic data in Table 1.  We made a figure clarifying schedule of the present study as Figure 1. We described the assay of serum BDNF in detail as follows.

In brief, 96-well microplates were coated with anti-BDNF monoclonal antibody and incubated at 4℃ for 18 hours. The plates were incubated in a blocking buffer for 1 hour at room temperature. The samples diluted with assay buffer by 100-times and BDNF standards were kept at room temperature under conditions of horizontal shaking for 2 hours, followed by washing with the appropriate washing buffer. The plates were incubated with antihuman BDNF polyclonal antibody at room temperature for 2 hours and washed with the washing buffer. The plates were then incubated with anti-IgY antibody conjugated to horseradish peroxidase for 1 hour at room temperature, and incubated in peroxidase substrate and tetramethylbenzidine solution to induce a color reaction. The reaction was stopped with 1 mol/L hydrochloric acid. The absorbance at 450 nm was measured with an Emax automated microplate reader. Measurements were performed in duplicates. The standard curve was linear from 0.5 ng/mL to 50ng/mL, and the detection limit was 5 ng/mL. Cross-reactivity to related neurotrophins (NT-3, NT-4, NGF) was less than 3%. Intra- and interassay coefficients of variation were about 5% and 7%, respectively. The recovery rate of the exogenous added BDNF in the measured serum samples was more than 95%. All measurements were performed in triplicate, and the average value was used as the measured value.

Our reply:
We did not perform the power analysis to determine the sample number. Thus, we added the point in the limitation as follows.

This study has important limitations. First is the flexible dose design, with the type and dosage of antidepressants left to the discretion of the attending physician. Second, the number of patients was small (n = 35), and additionally we did not perform the power analysis to determine the sample number. Third, there was no placebo group. Another limitation is the short clinical course of the patients, who were followed up for only 8 weeks after antidepressant administration. A large-scale study including a placebo group and longer duration follow up should be conducted in the future to overcome these limitations.

  1. Results:  Please present all figures in color and all statistical values in tabular form.

Our reply:

We remade the color figures, and all statistical values in tabular form.

  1. Discussion: The authors need to totally reorganize and fully expand this section. Starting with the summary of the previous section (Results), the authors need to develop discussion on the potential of this study complementing as the extension of the previous work, the implication of the findings of this study, how this study could facilitate future research, the ultimate goal, the challenge, the knowledge and the technology necessary to achieve this goal, the statement about this field in general, and finally the importance of this line of research. It is particularly important to present its limit and its merit, and its potential translation of this study to clinical practice.

Our reply:

We extended and arranged the Discussion as follows.

The results of the current study showed that serum BDNF concentrations increased significantly after 8 weeks in the paroxetine, escitalopram, and duloxetine response groups, but not after 2 or 4 weeks. However, serum BDNF concentrations did not increase at any time point in the non-response group. Previous studies have reported that serum BDNF concentrations increased during antidepressant treatment (19,20,33,34). However, the duration of the response and the type of drug used varied. In particular, selective serotonin receptor inhibitors (SSRIs) and serotonin-norepinephrine receptor inhibitors (SNRIs) have been shown to increase serum BDNF levels after 8 weeks of treatment (22,35). The current study examined serum BDNF concentrations after 8 weeks of paroxetine treatment and found that serum BDNF concentrations after 8 weeks significantly increased in the group of patients who had responded to paroxetine treatment, while serum BDNF concentrations after 4 weeks were unchanged. In contrast, there was no change in serum BDNF concentration in the group of non-responders. In the second set of 35 untreated patients, the serum BDNF concentration after 8 weeks of treatment with paroxetine, duloxetine, or escitalopram was significantly increased, while the serum BDNF concentration after 2 or 4 weeks of treatment was unchanged in the response group. In a study of sertraline, venlafaxine, and escitalopram, elevation in serum BDNF level was observed at 5 weeks and 6 months post-dose in the sertraline group and at 6 months post-dose in the venlafaxine group, whereas no change was observed in the non-responder group at any time point. In contrast, the escitalopram group showed no increase in serum BDNF after 6 months (36).

A decrease in serum BDNF levels in the early phase of SSRI treatment may be associated with a later SSRI response in adolescents with MD (37). A study reported that plasma BDNF was not significantly changed after 1-2 days of single ketamine administration compared to placebo, which does not support the hypothesis that ketamine treatment increases BDNF plasma levels in patients with MD (38). Another report demonstrated that BDNF was significantly elevated only 1 week following the first ketamine infusion in those classified as responders (39). No correlations were found between plasma BDNF levels and response to venlafaxine and paroxetine treatment at week 10 in patients with MD (40). Treatment with venlafaxine for 4 weeks decreased serum BDNF levels, whereas treatment with mirtazapine for 4 weeks increased serum BDNF levels in patients with MD (41). Treatment with mirtazapine for 12 weeks increased serum BDNF levels, which is associated with its response (42). Based on these findings, the relationship between antidepressants, duration of treatment, and treatment response is inconsistent and complicated.

We previously reported that early changes in serum BDNF levels (from week 0 and week 4) did not predict the response to treatment with SSRIs (43). A recent systematic review and network meta-analysis found a significant effect of antidepressants on increased BDNF levels [standardized mean difference (SMD) = 0.62; 95% confidence interval (CI) = 0.31-0.94, Z = 3.92, p < 0.0001] (25). Increases in BDNF levels over time were also associated with significant decreases in HAMD scores (SMD = 2.78, 95% CI = 2.31-3.26, Z = 11.57, p < 0.00001). The review also reported that SNRIs showed higher effect sizes than SSRIs (0.92 vs. 0.68). In addition, four antidepressants were analyzed separately for their role in increasing BDNF levels. Among these, only sertraline showed a significant increase in BDNF levels after treatment (SMD = 0.53, 95% CI = 0.13-0.93, Z = 2.62, p = 0.009), while venlafaxine, paroxetine, and escitalopram did not.

Furthermore, it has been reported that electroconvulsive therapy (ECT) also could alter serum BDNF levels in MD patients (44-48), but other reports did not produce the same findings (49,50). Repetitive transcranial magnetic stimulation (rTMS) also increases serum BDNF levels (51-54). Taken together, BNNP pathway is common pathway for antidepressants, ECT, and rTMS improve depressive symptoms, which is not still controversial.

In the current study, antidepressants generally had a significant effect on the increase in serum BDNF levels after 8 weeks. Our prospective study of serum BDNF concentrations in a relatively small number of patients demonstrated that antidepressant-responsive patients had the first significant increase after 8 weeks of treatment, while non-responders showed no change at any time point. These results are consistent with those of Zhou et al. (25) and did not contradict our previous report demonstrating that early changes in serum BDNF levels (from week 0 and week 4) did not predict the response to treatment with SSRIs (43). Another systematic review and meta-analysis reported that peripheral measurements of BDNF are inadequate predictors of treatment response in treatment-refractory MD patients (55). In our previous reports (56-58), catecholamine metabolites were altered after 4 weeks in the antidepressant response group, whereas BDNF was altered after 8 weeks in the antidepressant response group in the present results. These results suggest that changes in blood catecholamine metabolites precede changes in blood BDNF. Based on these findings, serum BDNF may be a candidate as a predictive factor for treatment response; however, it is difficult to predict treatment of MD simply from BDNF trajectory alone. Combining BDNF trajectory with other biomarkers and imaging findings may help to more accurately predict treatment response and prognosis.

This study has important limitations. First is the flexible dose design, with the type and dosage of antidepressants left to the discretion of the attending physician. Second, the number of patients was small (n = 35), and additionally we did not perform power analysis to determine the sample number. Third, there was no placebo group. Another limitation is the short clinical course of the patients, who were followed up for only 8 weeks after antidepressant administration. A large-scale study including a placebo group and longer duration follow up should be conducted in the future to overcome these limitations.

  1. Conclusion: I think that this section would benefit from a paragraph presenting some thoughtful as well as in-depth considerations by the authors as experts to convey the take-home message, as it is very descriptive but not enough theoretical as a conclusion should be. The authors should make their effort to explain the theoretical implication as well as the translational application of their research.

Our reply

We extended the Conclusion section as follows.

In patients with first-episode and drug-naive MD treated with antidepressants, serum BDNF concentrations in the treatment response group increased significantly only after 8 weeks but not after 2 or 4 weeks of treatment. In contrast, no change in the serum BDNF concentration was observed in the non-responder group at any time point.

The difference of the trajectory of serum BDNF levels between the responders and non-responders to antidepressants might be complicated, and must be further elucidated for each antidepressant, and we must also follow the trajectory until at remission of MD.

In any case, serum or plasma BDNF levels could not be a robust biomarker for the prediction of antidepressants in MD patients at present.

  1. References: I suggest presenting more references to support the authors’ argumentation needed to be addressed to improve the quality, its adequacy, and its readability of the manuscript. Typically, a paper like this cites 60-70 references.
    Overall, the manuscript contains one figure, four tables, and 25 references. I believe that the manuscript may carry important value in showing the difference in BDNF levels between treatment responders and treatment non-responders of MDD patients. I hope that, after careful revisions, the manuscript can meet the Journal’s high standards for publication.
    I declare no conflict of interest regarding this manuscript.

Our reply:

We inserted additional references, and total reference number is 60.

We hope the revised manuscript is suitable for publication in Biomedicine.

Best regards,

Reiji

Round 2

Reviewer 2 Report

The authors did an excellent job clarifying all the questions I have raised in my previous round of review. Currently, this paper entitled ‘Changes in serum brain-derived neurotrophic factor in patients with first-episode and untreated major depression’, is a well-written, timely piece of research that described the changes in serum brain-derived neurotrophic factor (BDNF) concentrations in inpatients and outpatients with major depression treated with antidepressants.

Overall, this is a timely and needed work. It is well researched and nicely written, therefore I believe that this paper does not need a further revision, therefore the manuscript meets the Journal’s high standards for publication.

I am always available for other reviews of such interesting and important articles.

Thank You for your work, Reviewer

Author Response

To Academic Editor

We re-revised our manuscript according to your suggestion. We hope the re-revised manuscript is suitable for publication.

Best regards,

Reiji
